# CoTDiff: Diffusion-based Image Synthesis with BiCoT

## Abstract

Recent advancements in Large Language Models (LLMs), Large Multi-Modal Models (LMMs), and text-to-image generation have significantly improved multimodal understanding and generation. However, a fundamental gap remains between human drawing processes and the iterative denoising mechanisms of existing diffusion-based models, leading to structural inaccuracies, prompt inconsistencies, and factual errors. To address this, we propose CoTDiff, a novel diffusion-based multi-stage image synthesis framework that integrates Chain-of-Thought (CoT) reasoning. This approach introduces two forms of CoT: textual CoT, where an LLM depicts the image layout based on the prompt, and diffusion CoT, which generates images in multiple stages—edge maps, grayscale images, and colorful images—mimicking the human drawing process.

CoTDiff leverages a feature insertion mechanism to harmonize these stages, effectively reducing conflicts and improving consistency. Empirical results demonstrate that CoTDiff outperforms existing text-to-image methods, particularly in complex tasks requiring accurate object counting and spatial control. By bridging the gap between human drawing and machine generation, CoTDiff offers a fresh perspective on integrating CoT into image synthesis and unlocks the latent potential of diffusion models to produce high-quality, detailed, and coherent images.

## 1 Introduction

Recent years have witnessed remarkable progress in Large Language Models (LLMs) Radford et al. (2019); Brown et al. (2020); Touvron et al. (2023) and Large Multi-modal Models (LMMs) Liu et al. (2024); Yang et al. (2025a); Ye et al. (2024a). Among them, DeepSeek DeepSeek-AI (2025) has brought the field of Natural Language Processing (NLP) to a new peak. A key technique that contributed to its success is Chain-of-Thought (CoT) Wei et al. (2023); Yao et al. (2023), which has significantly enhanced complex reasoning capabilities Wei et al. (2022) and performance in scientific and mathematical tasks Saikh et al. (2022).

This raises an intriguing question: can we combine CoT with image synthesis? To answer this question, we first need to understand the essence of CoT. Wei et al. Wei et al. (2023) defined it as a series of intermediate natural language reasoning steps that lead to a final output. In other words, CoT mimics the human reasoning process.

By analogy, CoT in image synthesis would mimic the human drawing process, in which a person first determines the image layout, then sketches the outline of objects in the image, followed by adding structure details, and finally adds color to the image.

In contrast, existing text-to-image models (T2Is) Liu et al. (2023); Xie et al. (2024); Ho et al. (2020), particularly diffusion models Song et al. (2020); Ramesh et al. (2021); Rombach et al. (2021), generate images through an entirely different process. They iteratively denoise Gaussian noise to produce the final image integrally. This mismatch between human drawing and diffusion generation can lead to structural errors, property mixing, prompt inconsistency, and factual inaccuracies.

Although the exact definition of CoT in diffusion models remains unclear, researchers are actively exploring this area Ye et al. (2024b); Yang et al. (2025b). For instance, T2I-R1 Jiang et al. (2025a) introduces two levels of CoT—semantic CoT and token CoT—to enhance prompt accuracy and consistency. Similarly, PARM Guo et al. (2025) scores intermediate images during generation to guide

the process toward human aesthetics. However, these methods still generate all image elements simultaneously, failing to close the gap between the diffusion process and human drawing. This gap often results in factual errors, structural inaccuracies, and lack of details.

To bridge this gap and unlock the potential of CoT in diffusion models, we propose a new framework: **CoTDiff**: a multi-stage diffusion-based image synthesis model with BiCoT. Unlike prior works Wu et al. (2025), our model employs two forms of CoT: textual CoT and diffusion CoT. Specifically, for the textual CoT, an LLM analyzes the image layout based on the textual prompt. For the diffusion CoT, a diffusion model generates the image in multiple stages: edge maps Xie & Tu (2015), grayscale images, and finally colorful images.

The inspiration for this approach comes from two observations. First, just as LLMs gain power by imitating human reasoning, image synthesis models can benefit from imitating human drawing. Second, empirical evidence shows that conditioning generation on intermediate representations like canny edges Zhang et al. (2023), scribbles Carrillo et al. (2023), or sketches Xu et al. (2024) produces higher-quality images than relying solely on textual descriptions Qin et al. (2023); Zhang et al. (2023). This suggests that generating the image in a single step does not fully leverage the potential of diffusion models.

Furthermore, we introduce a feature insertion mechanism to connect different stages of the image generation process. This effectively reduces generation intent conflicts between stages and improves the model's overall performance. Extensive experiments demonstrate that the CoTDiff model produces higher-quality images and excels in complex tasks, such as accurately generating a specified number of objects Binyamin et al. (2025); Kang et al. (2025) and controlling their positions.

The main contributions of this work are as follows:

- We propose a new form of CoT in diffusion models, bridging the gap between the diffusion process and human drawing, offering a fresh perspective on integrating CoT into image synthesis.

- A multi-stage image synthesis model with Bi-CoT is designed, which imitates human drawing and unlocks the potential of diffusion models to produce highly consistent and complex images.

- A feature insertion method is developed to harmonize different synthesis stages, effectively reducing generation intent conflicts and boosting model performance.

- Extensive experiments demonstrate the superior of the proposed CoTDiff model, which outperforms existing methods in text-to-image synthesis, especially on challenging tasks requiring accurate object counts and positions.

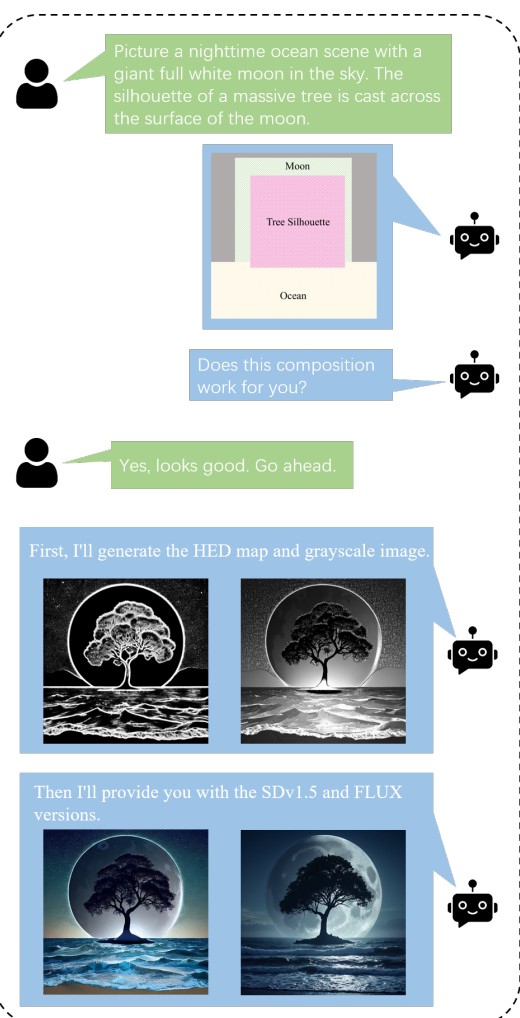

Figure 1: The image generation process of CoTDiff: given a textual description, CoTDiff first constructs the layout of the primary objects. Then sequentially generates a HED map, a grayscale image, and finally a color image.

## 2 RELATED WORK

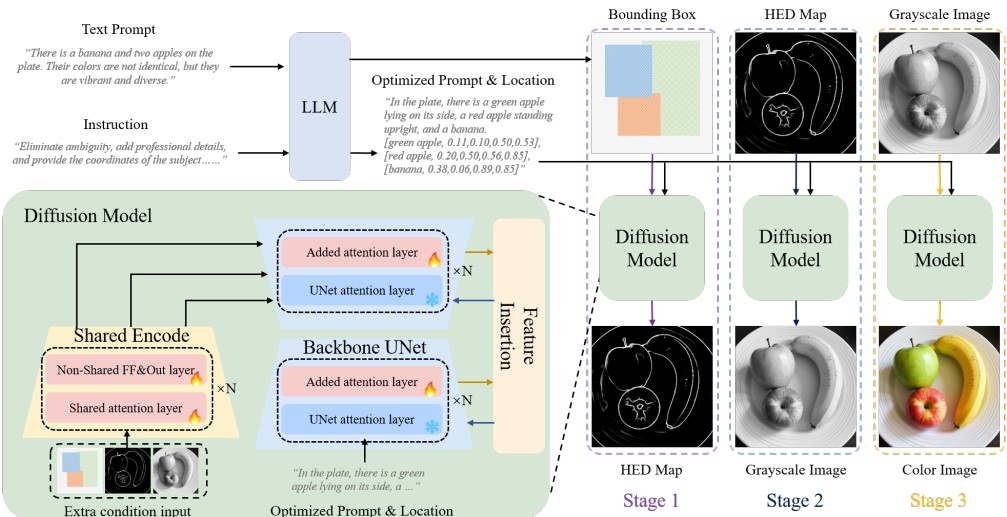

Figure 2: **The framework of CoTDiff**. Given a text prompt, an LLM first optimizes the prompt and provides the locations of the main objects. Then, a customized diffusion model generates the color image in three stages.

**Chain of Thought (CoT).** Chain of Thought (CoT) Wei et al. (2022) refers to a series of intermediate reasoning steps that significantly enhance the ability of LLMs to perform complex reasoning. In NLP, LLMs address problems where mapping an input $x$ directly to an output $y$ is non-trivial. The core idea of CoT is to introduce intermediate steps $z_1, \cdots, z_n$, where each $z_i$ is a coherent sequence that gradually bridges $x$ and $y$. These intermediate steps help the model break down complex tasks into simpler sub-problems. Extensive research Qiao et al. (2023) has shown that CoT, combined with few-shot in-context learning, can unlock the latent reasoning abilities of LLMs. Recently, Tree of Thought Yao et al. (2023) has been proposed to handle more complicated reasoning paths by exploring multiple branching thought processes.

**Diffusion Models.** In the field of image synthesis, research focus has shifted from GAN-based models Reed et al. (2016a;b) to diffusion models Chen et al. (2023; 2024); Peebles & Xie (2023), and further towards LMMs Xie et al. (2024; 2025). Despite their maturity and impressive image generation ability Labs (2024); Labs et al. (2025), diffusion models still struggle with complex tasks such as generating objects with precise numbers and positions, which cannot be resolved solely by scaling model parameters or relying on more complex textual prompts. Conceptually, diffusion models transform a simple known distribution, like Gaussian noise $\mathcal{N}$, into the complex distribution of real images $\mathcal{Z}$ through a sequence of transformations. These intermediate distributions $\mathcal{Z}_1, \cdots, \mathcal{Z}_n$ blend characteristics of both the noise and the target distribution. Unlike CoT in LLMs, which introduces explicit and interpretable intermediate reasoning steps, these intermediate transformations in diffusion models are implicit and lack semantic structure, limiting controllability and interpretability.

**Image Synthesis Models with CoT.** Applying CoT in language-based models is relatively straightforward. Researchers have also explored using CoT in LMMs for vision and video understanding tasks Dong et al. (2024b); Shao et al. (2024); Zheng et al. (2023). However, integrating CoT directly into image synthesis models remains an open challenge. One line of work leverages LMMs to produce tokenized images, as in T2I-R1 Jiang et al. (2025a). T2I-R1 applies semantic-level CoT to refine prompts, resolving ambiguity and adding detail, and token-level CoT to generate images patch by patch, ensuring that each new patch aligns coherently with previously generated patches, thereby improving consistency. Another approach, exemplified by PARM Guo et al. (2025), applies CoT-inspired reasoning directly within diffusion process. PARM parallelly explores multiple generation trajectories, scoring intermediate results and dynamically steering the generation process at each diffusion step to better align with human aesthetics.

Despite these promising efforts, existing approaches still generate all image elements simultaneously and do not fully mimic the human drawing process, which typically follows a multi-stage strategy: sketching, refining, and coloring. This gap motivates our work to explore a more human-like, multi-stage image synthesis approach with BiCoT.

## 3 METHOD

In this section, we first provide an overview of the diffusion process and formalize the task of integrating CoT into diffusion models. We then introduce our proposed BiCoT framework, which combines Textual CoT and Diffusion CoT. Finally, we describe a feature insertion strategy designed to bridge different diffusion stages and enhance consistency during image generation.

### 3.1 PRELIMINARY

Diffusion models can be intuitively understood as a process similar to how ink disperses in water—from a concentrated state to a diffused one. In Denoising Diffusion Probabilistic Models (DDPMs) Ho et al. (2020) and Latent Diffusion Models (LDMs) Rombach et al. (2021), this process is known as the *forward process* or the *diffusion process*, representing an increase in entropy. On the contrary, the *reverse process*, an entropy-reducing process, which is learned during training, gradually removes noise to recover the data.

During training, the model learns to predict noise $\epsilon_t$ added at timestep $t$, and the training objective is the denoising loss:

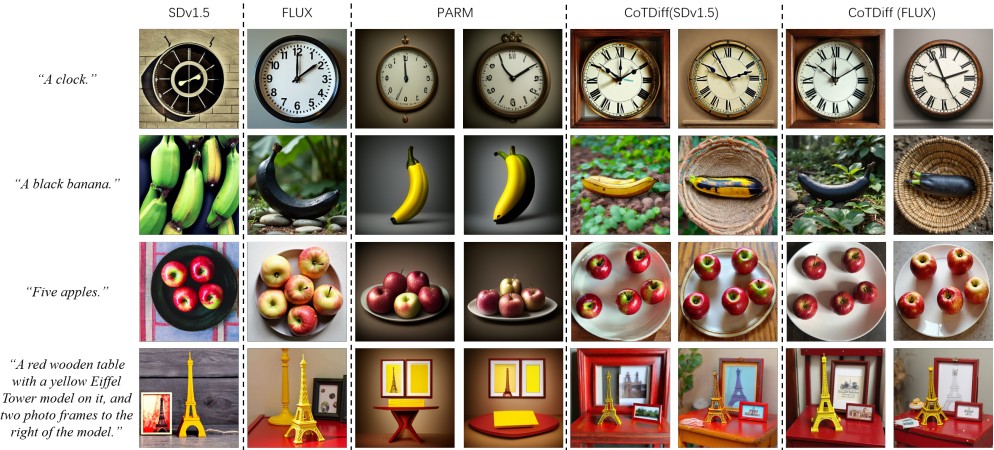

Figure 3: Visualization of comparative experimental results.The first row demonstrates that CoTDiff enhances the level of detail in the generated images and improves the accuracy of word rendering.The second row illustrates the reflective role of FLUX in refining CoTDiff's outputs.The third and fourth rows highlight the superiority of our model in tasks involving object counting, attribute binding, and spatial positioning.

$$\mathcal{L} = \mathbb{E}_{x,\epsilon_t \sim \mathcal{N}(0,\mathbf{I}),t} \left[ ||\epsilon_t - \epsilon_\theta(x_t, t, \varphi(Y))||_2^2 \right], \tag{1}$$

where $\theta$ represents the model parameters, $x$ is the latent image representation, $Y$ is the textual description, and $\varphi(\cdot)$ is the text encoder.

Applying CoT to diffusion remains an emerging challenge. Analogous to its role in LLMs, CoT in diffusion aims to break down the transformation from Gaussian distribution $\mathcal{N}$ to real image distribution $\mathcal{Z}$ into a sequence of interpretable intermediate distributions $\mathcal{Z}_1, \cdots, \mathcal{Z}_n$—each representing a meaningful stage toward final image synthesis.

## 3.2 TEXTUAL CoT

Textual CoT refers to the reasoning and semantic planning phase that precedes image generation. Just as an artist interprets a prompt, refines the description, and mentally plans the composition, our model uses an LLM to perform similar operations.

As shown in Fig.2, Given a short or ambiguous prompt, the LLM follows the instruction, via in-context learning Dong et al. (2024a); Zhou et al. (2023), and generates a detailed and clarified description along with layout guidance in the form of bounding boxes, defined by left-top and right-bottom coordinates.

Textual CoT offers two key benefits:

*Disambiguation and enhancement*: For example, a prompt like "a girl in red and a luminophor" is transformed into "A girl in a red hooded cloak kneels on the ground, gently holding a softly glowing yellow luminophor", which is more descriptive and aligned with human aesthetics.

*Layout grounding*: Providing spatial constraints through bounding boxes gives diffusion models more direct guidance, especially since they often struggle with interpreting locational nouns.

## 3.3 DIFFUSION CoT

While CoT is well defined in language models, its role in diffusion remains unclear. Prior methods such as PARM Guo et al. (2025) attempt to leverage intermediate results during denoising by decoding them and scoring possible continuations. However, such approaches resemble heuristic search and lack semantic structure.

We propose a novel Diffusion CoT strategy, inspired by the empirical observation that conditioning image generation on intermediate representations—such as canny edges, scribbles, or sketches—yields higher quality results than textual prompts alone Qin et al. (2023); Zhang et al. (2023).

We define a semantically meaningful CoT sequence:

$\mathcal{Z}_1$: HED edge map

$\mathcal{Z}_2$: Grayscale image

$\mathcal{Z}_3$: Color image

These are generated sequentially:

Generate $\mathcal{Z}_1$ (HED) conditioned on description $Y$ and the generated bounding box $B$ in Textual CoT.

Generate $\mathcal{Z}_2$ (grayscale) extra conditioned on $\mathcal{Z}_1$.

Generate $\mathcal{Z}_3$ (color) extra conditioned on $\mathcal{Z}_2$.

As shown in Fig.2, this is implemented using a parameter-shared ControlNet, where attention block parameters are shared across stages, while feedforward and output blocks remain stage-specific. The generation process is:

$$\mathcal{Z}_i = \phi_{\theta_s, \theta_i}(\mathcal{Z}_{i-1}, \varphi(Y), B), \tag{2}$$

where $\phi(\cdot)$ is the image synthesis process, $\theta_s$ are shared parameters, $\theta_i$ are stage-specific parameters, and $\varphi(\cdot)$ is the text encoder.

To insert the bounding box information into the UNet backbone, we follow Li et al. (2023) and add a gated cross-attention block. We encode local text and bounding box as grounding tokens and use cross-attention to inject locational information.

## 3.4 FEATURE INSERTION

Although the CoT stages are now sequential, they are still functionally isolated. To better integrate information across stages and reduce semantic conflict, we introduce a feature insertion strategy

inspired by previous work on long-term feature banks Wu et al. (2019); He et al. (2025); Liang et al. (2024).

We modify the attention blocks in the UNet to incorporate external features. Given a latent representation $x_i$ at step $i$, we project it to query-key-value tokens:

$Q_t \in \mathbb{R}^{n \times d}$: current queries, $n$ denotes the number of tokens, and $d$ denotes the dimension of the latent.

$K_t, V_t \in \mathbb{R}^{n \times d}$: current keys and values

$K_s, V_s \in \mathbb{R}^{m \times d}$: stored keys and values from prior stages, $m$ denotes the size to stored features

We then define an extended attention operation:

$$\text{Attn} = \text{softmax}\left(\frac{Q_t \cdot [K_t, K_s]^\top}{\sqrt{d}}\right)[V_t, V_s], \tag{3}$$

where $[\cdot]$ denotes concatenation.

As shown in Fig.**??**, to initialize and update stored features, we consider two naïve strategies:

*Queue-based*: Maintain a fixed-size queue, dropping the oldest features when full.

*Averaging-based*: When $n = m$, update stored features via averaging:

$$K_s = \frac{K_s + K_{\text{new}}}{2}, \ V_s = \frac{V_s + V_{\text{new}}}{2}. \tag{4}$$

While efficient, this may lead to semantic drift if features at corresponding positions are not aligned.

To address this, we adopt a token merging method Bolya et al. (2023). For a new value token $v_{\text{new},k}$, we compute cosine similarity:

$$\text{sim}(v_{s,j}, v_{\text{new},k}) = \frac{v_{s,j} \cdot v_{\text{new},k}}{|v_{s,j}||v_{\text{new},k}|} \quad j = 1, 2, \cdots, m, \tag{5}$$

and merge it with the most similar stored token $l$:

$$v_{s,l} \leftarrow \frac{v_{s,l} + v_{\text{new},k}}{2}. \tag{6}$$

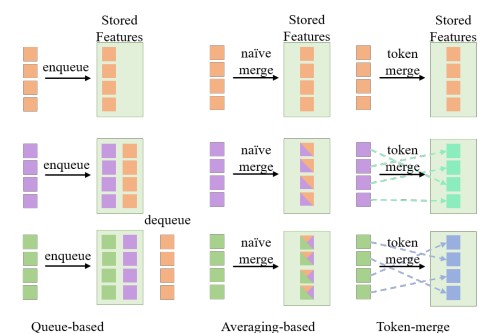

Figure 4: Three feature insertion strategies. The queue-based method, which has high memory consumption and slow reasoning; the averaging-based method, which ignores feature alignment; and the token-merge method, which merges the most similar features.

This method updates the top-$n$ stored tokens based on similarity, reducing storage while preserving semantic alignment.

## 3.5 MODEL OPTIMIZATION

The LLM is frozen due to in-context learning in textual CoT. During diffusion CoT training, we freeze the parameters of the backbone UNet and optimize only the added cross-attention layers and the parameter-shared ControlNet. Each training sample is a quaternion consisting of the target image $\mathcal{Z}_i$, the conditioning image $\mathcal{Z}_{i-1}$, the textual description $Y$, and the bounding box $B$.

The optimization process follows a similar strategy to Stable Diffusion (SD). Specifically, given a target image $\mathcal{Z}_i$, we first encode it into the latent space using the encoder of a pre-trained autoencoder. We then randomly sample a timestep $t \in [0, T]$ and add Gaussian noise to obtain the noisy latent $\mathcal{Z}_{i,t}$.

Since multiple image pairs $(\mathcal{Z}_i, \mathcal{Z}_{i-1})$ exist across different stages (e.g., HED-to-gray, gray-to-color), we randomly sample different stage groups within each batch during training to ensure balanced learning across stages.

The model is trained to predict the noise $\epsilon_t$ given the conditioning image, text, and layout constraints. The training objective is formulated as:

$$\mathcal{L} = \mathbb{E}_{\mathcal{Z}_{i,t},t,\epsilon_t \sim \mathcal{N}(0,\mathbf{I})} \left[ ||\epsilon_t - \epsilon_\theta(\mathcal{Z}_{i-1}, t, \varphi(Y), B)||_2^2 \right], \tag{7}$$

where $\epsilon_\theta$ is the model's predicted noise, $\varphi(Y)$ is the text embedding of the description $Y$, and $B$ represents the bounding box layout.

| Model | Aesthetic Score | Parameters / B | Speed / s |
|---|---|---|---|
| SDv1.5 | 4.94 | 1.1 | 4.1 |
| FLUX | 5.93 | 16.9 | 28.5 |
| PARM | 3.90 | 9.6 | 101.3 |
| CoTDiff (SDv1.5) | 5.09 | 1.8 | 7.6 |
| CoTDiff (FLUX) | 5.96 | 18.7 | 36.0 |

Table 1: Performance comparison on the GenEval test datasets based on aesthetic score, model size, and inference speed. The state-of-the-art (SOTA) model, PARM, achieves the slowest inference speed.

| Model type | Model | Single object | Two object | Counting | Colors | Position | Attribute binding | Overall |
|---|---|---|---|---|---|---|---|---|
| Auto-Regressive | LlamaGen | 0.71 | 0.34 | 0.21 | 0.58 | 0.07 | 0.04 | 0.32 |
| | LWM | 0.93 | 0.41 | 0.46 | 0.79 | 0.09 | 0.15 | 0.47 |
| | SEED-X | 0.97 | 0.58 | 0.26 | 0.80 | 0.19 | 0.14 | 0.49 |
| | show-o | 0.95 | 0.52 | 0.49 | 0.82 | 0.11 | 0.28 | 0.53 |
| | Janus-Pro | 0.99 | 0.89 | 0.59 | 0.90 | 0.79 | 0.66 | 0.80 |
| | PARM(show-o) | 0.99 | 0.86 | 0.67 | 0.84 | 0.66 | 0.64 | 0.77 |
| | PARM(Janus-Pro) | 1.00 | 0.95 | 0.80 | 0.93 | 0.91 | 0.85 | 0.91 |
| Diffusion | minDALL-E | 0.73 | 0.11 | 0.12 | 0.37 | 0.02 | 0.01 | 0.23 |
| | CLIP retrieval | 0.89 | 0.22 | 0.37 | 0.62 | 0.03 | 0.00 | 0.35 |
| | SDv1.5 | 0.97 | 0.38 | 0.32 | 0.72 | 0.04 | 0.08 | 0.42 |
| | PixArt-$\alpha$ | 0.98 | 0.50 | 0.44 | 0.80 | 0.08 | 0.07 | 0.48 |
| | SDv2.1 | 0.98 | 0.51 | 0.44 | 0.85 | 0.07 | 0.17 | 0.50 |
| | DALL-E 2 | 0.94 | 0.66 | 0.49 | 0.77 | 0.10 | 0.19 | 0.52 |
| | SD-XL | 0.98 | 0.74 | 0.39 | 0.85 | 0.15 | 0.23 | 0.55 |
| | IF-XL | 0.97 | 0.74 | 0.66 | 0.81 | 0.13 | 0.35 | 0.61 |
| | SD 3 | 0.98 | 0.74 | 0.63 | 0.67 | 0.34 | 0.36 | 0.62 |
| | FLUX | 0.99 | 0.84 | 0.71 | 0.78 | 0.20 | 0.47 | 0.67 |
| | CoTDiff (SDv1.5) | 0.98 | 0.84 | 0.72 | 0.70 | 0.65 | 0.11 | 0.67 |
| | CoTDiff (FLUX) | 0.99 | 0.85 | 0.73 | 0.79 | 0.63 | 0.49 | 0.75 |

Table 2: Performance comparison on the GenEval benchmark. Both versions of CoTDiff achieve significant improvements over their respective base models.

## 4 EXPERIMENT

In this section, we describe the experimental setup, including the baseline model, hyperparameter settings, datasets, and evaluation metrics. We then present the main results of our proposed model, CoTDiff, and compare its performance with state-of-the-art methods. Finally, we conduct ablation studies to analyze the individual contributions of the Diffusion CoT and Feature Insertion strategies.

### 4.1 EXPERIMENTAL SETUP

**Training Details** We train CoTDiff for 6,000 steps with a batch size of 32, using the Adam optimizer Kingma & Ba (2017) and a learning rate of $1 \times 10^{-4}$. Input and conditioning images are resized to $512 \times 512$ pixels. Our backbone model is based on Stable Diffusion v1.5 (SDv1.5), with DeepSeek-R1 used as the large language model for Textual CoT. Due to the limited generation quality of

SDv1.5, we also train an enhanced version of CoTDiff using FLUX as an additional fourth-stage module for image super-resolution and restoration. Training is conducted on 4 NVIDIA A800 GPUs (80 GB), and costs almost two days.

**Datasets** Our primary experiments are conducted on the COCO2017 training dataset Lin et al. (2015). To improve image quality, we augment the dataset with synthetic images generated by FLUX Labs (2024), using COCO2017 captions as prompts. HED edge maps for Diffusion CoT are obtained using the method of Zhang et al. (2023). For evaluation, we main test our model in datasets generated by GenEval benchmark Ghosh et al. (2023).

**Feature Insertion Settings** To reduce memory consumption and accelerate inference, we set the stored feature size to $m = 1$. When generating colorful images from grayscale inputs, the Control-Net conditioning scale is set to 0.375; for all other cases, it is set to 1.0. For queue-based feature insertion, the maximum queue size is set to 3.

**PARM** We use the default settings of PARM, and it is worth noting that the search number is set to 20. This means PARM initiates 20 diffusion flows during a single image generation process.

| Model | Single object | Two object | Counting | Colors | Position | Attribute binding | Overall |
|---|---|---|---|---|---|---|---|
| SDv1.5 | 0.97 | 0.38 | 0.32 | 0.72 | 0.04 | 0.08 | 0.42 |
| Diffusion CoT | 0.98 | 0.73 | 0.72 | 0.70 | 0.59 | 0.10 | 0.63 |
| Queue-based | 0.98 | 0.72 | 0.64 | 0.66 | 0.61 | 0.07 | 0.61 |
| Averaging-based | 0.98 | 0.72 | 0.74 | 0.66 | 0.63 | 0.12 | 0.65 |
| Token-merge | 0.98 | 0.84 | 0.72 | 0.70 | 0.65 | 0.11 | 0.67 |

Table 3: Ablation study on the GenEval benchmark. The first two rows demonstrate that Diffusion CoT provides significant improvement in handling complex tasks, while the last three rows highlight the superiority of the Token-merge method.

**Evaluation Metrics and Benchmark** We evaluate CoTDiff using the following metrics:

*GenEval* Ghosh et al. (2023): An object-centric evaluation framework that assesses compositional properties such as object co-occurrence, spatial layout, object count, and color accuracy. It is particularly effective in testing compositional reasoning and grounding in generative models.

*Aesthetic Score* discus0434 & Goswami (2025): A learned predictor that scores images on a scale from 1 to 10, with higher scores indicating more visually pleasing and human-aligned aesthetics.

### 4.2 MAIN RESULTS

We compare CoTDiff with leading text-to-image generation models, including both diffusion-based and autoregressive methods, using the GenEval benchmark. We also report aesthetic scores and inference speed in Table 1, and provide qualitative results in Table 2.

Our model demonstrates significant improvements over the SDv1.5 baseline. Specifically, as shown in Table 2, CoTDiff (SDv1.5) increases the overall GenEval score by 0.25, corresponding to a 60% relative improvement. In the "Single Object" and "Two Objects" categories, CoTDiff improves by 0.01 and 0.46 (1% and 121% relative improvement) respec-

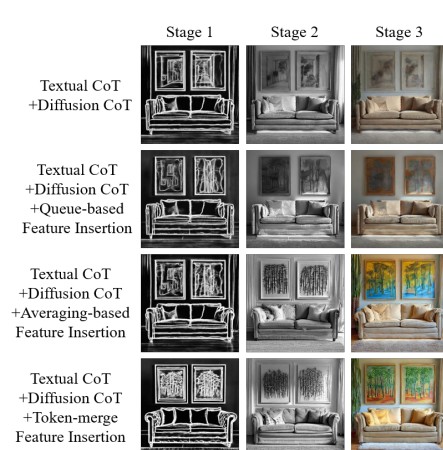

*"A sofa with two paintings hanging on the wall above it."*

Figure 5: Visualization of ablation study results. When implementing BiCoT with averaging-based and token-merge strategy, CoTDiff effectively integrate feature information and resolve inter-stage inconsistencies, with the latter yielding superior results.

tively, indicating that our CoT-enhanced diffusion framework effectively reduces object omission and enhances multi-object composition accuracy.

These improvements stem from two key innovations:

Textual CoT introduces explicit spatial descriptions and object-level reasoning, enabling the model to better understand and follow compositional prompts.

Multi-stage Diffusion Generation decouples layout planning from detail rendering. In the early stage, the model generates HED maps from bounding boxes and prompts, allowing it to determine object placement without being constrained by appearance features. Later stages refine object details and color.

In the "Counting" and "Position" categories, CoTDiff improves by 0.40 and 0.61 (125% and 1525% relative improvement) respectively. These gains are largely due to Textual CoT, which alleviates the counting and positioning pressure that is quite challenging for text encoders, even for powerful encders like T5 Raffel et al. (2020) or other LLMs.

However, gains in the "Color" and "Attribute Binding" categories are modest. This is partly due to the GenEval benchmark's use of random target colors from a fixed set, leading to out-of-distribution prompts (e.g., "black watermelon"). SDv1.5 struggles with such unrealistic cases, limiting its performance. Fortunately, just as CoT improves reasoning in LLMs, reflection Jiang et al. (2025b) can help correct previous generation errors. Our enhanced version, CoTDiff (FLUX), leverages FLUX's superior foundational capabilities to reduce color and attribute mismatches.

As shown in Table 1, we report the aesthetic scores, parameter counts, and inference speeds for various models. For aesthetic scores, both versions of CoTDiff demonstrate clear improvements over their respective baselines, and CoTDiff (SDv1.5) offers a competitive balance of generation quality, speed, and model size.

As shown in Fig.3, CoTDiff achieves significant performance improvements across various visual understanding tasks, particularly in image detail synthesis, counting, positional reasoning, and attribute association. While PARM also shows comparable improvements on individual tasks by generating multiple images simultaneously—albeit with considerably longer generation times—our experimental results (as shown in the bottom row) highlight its limitations in handling complex, multi-faceted generation scenarios that require integrated task processing.

### 4.3 ABLATION STUDIES

In this section, we investigate the effects of the Feature Insertion strategy and the multi-stage generation process. Quantitative and qualitative results are provided in Table 3.

Comparing the base SDv1.5 with the model incorporating Diffusion CoT, we observe significant gains in two object object, counting, spatial positioning and attribute binding.

Furthermore, we evaluate three different Feature Insertion strategies. Among them, the Token-Merge method delivers the best overall performance, striking an effective balance between accuracy and computational efficiency. As shown in Fig.5, the token-merge approach maintains semantic consistency across stages while eliminating interference between heterogeneous feature representations in different stages.

## 5 CONCLUSION

Currently, there is a significant gap between the human-like reasoning process required for high-quality image synthesis and the capabilities of existing diffusion-based models. To bridge this gap, we propose CoTDiff, a novel multi-stage image synthesis framework that integrates textual and diffusion CoT reasoning to mimic the human drawing process. A feature insertion mechanism is introduced to harmonize the stages of image generation, effectively improving consistency and reducing generation intent conflicts. By imitating human reasoning and drawing, CoTDiff extends the theoretical and practical potential of diffusion models, offering a fresh perspective on multi-stage image synthesis. Extensive experiments demonstrate the effectiveness of CoTDiff, showcasing its superiority in generating high-quality, consistent images.

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

## A APPENDIX

We used large language models (LLMs) to assist in polishing and refining the sentences in this paper. The content, ideas, and analyses are entirely our own, with LLMs employed solely for language enhancement purposes.

