# OpenReview forum: "CoTDiff: Diffusion-based Image Synthesis with BiCoT"
_ICLR.cc/2026/Conference — ICLR 2026 Conference Withdrawn Submission_

### Official Review · Reviewer_U1tR · 2025-10-15

**Soundness:** 2
**Presentation:** 1
**Contribution:** 2
**Rating:** 2
**Confidence:** 5

**Summary:**

CoTDiff is a multi-stage text-to-image method that adds a “textual CoT” step to predict layout boxes and a “diffusion CoT” pipeline that generates edge → grayscale → color, with a feature-insertion module to pass information across stages.

**Strengths:**

- Simple grounding with gated cross-attention and stored features is easy to plug into SD-1.5.
- GenEval results and ablations indicate some gains and that token-merge helps.

**Weaknesses:**

- The main boost may come from explicit layout boxes, not real “reasoning.” No ablation that removes the textual CoT while keeping diffusion CoT.
- Evaluation is narrow. It focuses on GenEval. It lacks more baselines and human studies.
- Novelty is modest. Prior work already uses ControlNet, layout grounding, and feature banks. This looks like a pipeline of known parts.
- Paper quality is weak. Writing is rough; some plots are hard to read.
- Method details are thin. Data scale, LLM prompts, error stats for box prediction, and failure cases are missing.

**Questions:**

- How much do scores drop if you remove the textual CoT and keep the diffusion stages and feature insertion? Please report a full ablation.
- What is the speed and memory cost for different feature-insertion sizes m?

---

### Official Review · Reviewer_7CyX · 2025-10-24

**Soundness:** 2
**Presentation:** 1
**Contribution:** 2
**Rating:** 2
**Confidence:** 4

**Summary:**

This paper proposes CoTDiff, a diffusion-based image synthesis framework that introduces Chain-of-Thought (CoT) reasoning into the image generation process. The model integrates two types of CoT: textual CoT, where a large language model refines prompts and layouts, and diffusion CoT, where the generation proceeds in multiple stages (edge → grayscale → color). A feature insertion mechanism is introduced to connect these stages.

**Strengths:**

1. The idea of mimicking human drawing stages (outline → detail → color) is intuitively appealing and helps organize the diffusion process more structurally.

2. The paper includes comparisons with both autoregressive and diffusion-based baselines.

**Weaknesses:**

1. The combination of comments does not yield a clear theoretical or methodological breakthrough beyond prior works.

2. The manuscript has numerous typesetting and referencing issues.

3. The loss equations (e.g., Eq. (1) and Eq. (7)) are typeset in a non-standard format.

4. Improvements are mostly moderate and limited to certain categories.

Overall, the paper’s presentation quality could be improved, as there are noticeable formatting and referencing issues that affect readability.

**Questions:**

1. See in W1.

2. Fix all figure references and reformat all mathematical equations.

3. Provide an ablation or visualization that demonstrates why multi-stage CoT reasoning leads to better interpretability.

The paper presents a moderately interesting idea but lacks originality and suffers from serious formatting and presentation issues. With improved writing, clearer novelty positioning, and a more rigorous theoretical grounding, it could be a stronger submission in future iterations.

**Details Of Ethics Concerns:**

No.

---

### Official Review · Reviewer_1ucZ · 2025-10-31

**Soundness:** 3
**Presentation:** 1
**Contribution:** 2
**Rating:** 2
**Confidence:** 4

**Summary:**

The work proposes to incorporate the CoT idea into the task of image generation from text. Different from previous methods, the new method generates an image from three steps: first generate a canny image from text, and then a grayscale image from the canny image, and then the colored image from the grayscale image. The method is significantly different from previous work and shows some performance improvement in the text-to-image generation task.

**Strengths:**

The work proposes a new type of CoT approach to image generation. In particular, it generates the image through the canny image and the grayscale image. As in the analysis, the canny graph seems to be a proper intermediate step for text-to-image generation.

**Weaknesses:**

The paper is poorly written and not really finished. There are several writing issues making the paper hard to read.
1. Some notations are not well explained. What's B in (2)?
2. The section 3.4 is not readable. What are Q and K matrices? I don't see where they are from.
3. The LLM-style writing of Sections 3.3 and 3.4 are very hard to read. With many short sentences and itemizations, it is hard to see the rationale behind the design.
4. Many citations are not correctly formatted. The reference to a figure in line 286 is missing.

The performance in many categories only has marginal performance over the baseline. For example, on the categories of Single object, Two object, counting, and Attribute binding, there is only marginal performance improvement.

Overall, this paper doesn't seem to be completed, even though the idea is novel.

**Questions:**

No questions.

**Details Of Ethics Concerns:**

The model proposes a new method that aims to improve Text-to-Image generation performance. Image generative models can be used to generate harmful content. The work should have a discussion about the potential concern.

---

### Official Review · Reviewer_RGJD · 2025-11-01

**Soundness:** 2
**Presentation:** 2
**Contribution:** 2
**Rating:** 2
**Confidence:** 4

**Summary:**

The authors CoTDiff as a method to enhance the generation quality of diffusion models (including flow-matching models). Mentioning that the previous methods that utilize CoT to boost the generation quality for the image generation task, the authors propose a mechanism that consists of two chains, where one is based on CoT occurring with the LLM and the second one is occurring within diffusion models as imaging steps. In the same way that CoT for text imitates the thought process with each generated text across thought steps, authors claim that over a proposed chain for diffusion (HED to grayscale to RGB), the same process can be imitated for image generation. Over the experiments conducted, the proposed method leads to quantifiable improvements across several generation tasks.

**Strengths:**

- The authors introduce a thought process for images, that is shown to be effective for the image generation task.
- The reported experiments demonstrate gains for both aesthetics improvements and task success across the included benchmark.
- The proposed method is simple and easily applicable to different generation backbones.
- Experiments show the effectiveness of the approach in different generation paradigms (autoregressive and diffusion based generation).

**Weaknesses:**

- Undefined reference in line 286.
- Since the method defends that image-CoT is a main factor that improves image generation, the expectation is every step of the chain to have a significant effect. This is not shown by any ablations. From the qualitative and quantitative results we see that there are performance improvements, but it is not clear if it is from multi-stage chain for diffusion-CoT. This is one factor that is crucial to be ablated.
- The method claims that having both text-CoT and diffusion-CoT is the core of the method design and a factor that improves the generation performance. Over the experiments, this difference is not investigated. While both of the CoT variants for generation may be found effective, the paper does not provide any insights on what is their corresponding effect.
- Upon the mentioned points on missing ablations, the main factor that improves the image quality seems more like the layout generation and its injection. To remove such questions, authors should provide necessary experiments over the revision.
- The provided experiments only compare with the base models, ideally comparisons with similar efforts that utilize LLMs to boost the generation quality should be a part of the experiments.

**Questions:**

- How did the CoT sequence is finalized, is the sequence of HED to grayscale and grayscale to RGB is the optimal sequence for the thought sequence?
- What are the impact of each stage occurring in diffusion-CoT?

---

### Note · Authors · 2025-11-12

**Comment:**

We take the reviewer's suggestions into account and realize that the experiment needs improvement and the paper requires significant changes.

**Withdrawal Confirmation:**

I have read and agree with the venue's withdrawal policy on behalf of myself and my co-authors.